# Genome Characterization and Infectivity Potential of Vibriophage-ϕLV6 with Lytic Activity against Luminescent Vibrios of *Penaeus vannamei* Shrimp Aquaculture

**DOI:** 10.3390/v15040868

**Published:** 2023-03-28

**Authors:** Manikantha Benala, Murugadas Vaiyapuri, Visnuvinayagam Sivam, Karthika Raveendran, Mukteswar Prasad Mothadaka, Madhusudana Rao Badireddy

**Affiliations:** 1Visakhapatnam Research Centre of ICAR-Central Institute of Fisheries Technology (ICAR-CIFT), Visakhapatnam 530003, India; 2Department of Microbiology and FST, School of Science, GITAM, Visakhapatnam 530045, India; 3ICAR-Central Institute of Fisheries Technology (ICAR-CIFT), Willingdon Island, Cochin 682029, India

**Keywords:** vibriophage, whole genome sequence, comparative genomics, *Vibrio harveyi*, luminescent vibrios, *Penaeus vannamei*, phage therapy

## Abstract

Shrimp aquaculture, especially during the hatchery phase, is prone to economic losses due to infections caused by luminescent vibrios. In the wake of antimicrobial resistance (AMR) in bacteria and the food safety requirements of farmed shrimp, aqua culturists are seeking alternatives to antibiotics for shrimp health management, and bacteriophages are fast emerging as natural and bacteria-specific antimicrobial agents. This study analyzed the whole genome of vibriophage-ϕLV6 that showed lytic activity against six luminescent vibrios isolated from the larval tanks of *P. vannamei* shrimp hatcheries. The Vibriophage-ϕLV6 genome was 79,862 bp long with 48% G+C content and 107 ORFs that coded for 31 predicted protein functions, 75 hypothetical proteins, and a tRNA. Pertinently, the vibriophage-ϕLV6 genome harbored neither AMR determinants nor virulence genes, indicating its suitability for phage therapy. There is a paucity of whole genome-based information on vibriophages that lyse luminescent vibrios, and this study adds pertinent data to the database of *V. harveyi* infecting phage genomes and, to our knowledge, is the first vibriophage genome report from India. Transmission electron microscopy (TEM) of vibriophage-ϕLV6 revealed an icosahedral head (~73 nm) and a long, flexible tail (~191 nm) suggesting siphovirus morphology. The vibriophage-ϕLV6 phage at a multiplicity of infection (MOI) of 80 inhibited the growth of luminescent *V. harveyi* at 0.25%, 0.5%, 1%, 1.5%, 2%, 2.5%, and 3% salt gradients. In vivo experiments conducted with post-larvae of shrimp showed that vibriophage-ϕLV6 reduced luminescent vibrio counts and post-larval mortalities in the phage-treated tank compared to the bacteria-challenged tank, suggesting the potentiality of vibriophage-ϕLV6 as a promising candidate in treating luminescent vibriosis in shrimp aquaculture. The vibriophage-ϕLV6 survived for 30 days in salt (NaCl) concentrations ranging from 5 ppt to 50 ppt and was stable at 4 °C for 12 months.

## 1. Introduction

Aquaculture, the farming of aquatic animals, significantly contributes to meeting the global demand for animal protein and generates large-scale employment opportunities for millions of people worldwide [1]. Shrimp farming is a major aquaculture activity in several countries, and farmed shrimp have a significant trade value across the globe. However, bacterial infections caused by vibrio species are extremely hazardous for sustainable shrimp aquaculture. Vibrio species such as *V. parahaemolyticus*, *V. harveyi*, *V. alginolyticus*, *V. campbelli*, *V. penaeicida*, *V. splendidus*, *V. fluvialis*, and *V. tubiashii* cause infections in aquatic animals. *V. harveyi* is the most important bacterial pathogen of penaeid shrimp that causes mortalities, especially in larval shrimp, and causes huge losses to shrimp aquaculture [2,3,4,5], which have been reported in several countries [6,7,8]. *V. harveyi* is the etiological agent of luminescent vibriosis that causes the affected aquatic animals to glow in the dark, and the pathogenicity mechanisms involve bacterial LPS and proteases [9]. Luminescent *V. harveyi* was detected in shrimp hatcheries [8,10] but was more frequently recovered from hatcheries affected by luminescent bacterial disease [11]. Use of antibiotics [12,13] for the control of vibrio bacteria has led to the emergence of antimicrobial resistance in vibrio species [14], and AMR vibrios were reported to cause mortalities in *Penaeus monodon* shrimp larvae [8] and *P. vannamei* shrimp [15]. Moreover, the use of antibiotics in food-producing animals has implications for food safety.

Alternatives to antibiotics for aquatic animal-health management are being increasingly pursued, and in this regard, bacteriophages, the viruses that kill bacteria, are being actively revisited as biocontrol agents in aquaculture [16]. Lytic vibriophages are bacteriophages that have the ability to lyse Vibrio bacteria, are being investigated as potential biocontrol agents for use in aquaculture [17,18]. Phage therapy employing vibriophages has been researched and found to be effective in controlling vibrio infections in aquaculture [18,19,20], and phages were reported to be successful in inhibiting the growth of *Vibrio parahaemolyticus* [21,22,23], *Vibrio coralliilyticus*, *Vibrotubiashii* [24], *Vibrio anguillarum* [25], *Vibrio alginolyticus* [26,27], *Vibrio campbelli* [28], and *Vibrio harveyi* [17,29,30,31,32,33] in aquatic animals. Phages with lytic activity against *V. harveyi* were isolated from *P. monodon* shrimp farms and hatchery waters [11,29,30,34,35]. Further, bacteriophages active against *V. harveyi* were reported to be ineffective against microorganisms beneficial for aquaculture, such as probiotic bacteria and nitrifying bacteria [36]. However, complete genomic characterization of the bacteriophages vis-à-vis antibiotic-resistance genes (ARGs) and virulence genes is a pre-requisite prior to their selection for application as biocontrol agents [37,38,39,40]. The present study reports the isolation, genomic characterization, and infectivity assessment of vibriophage-ϕLV6 with lytic activity against luminescent *V. harveyi* for potential application in shrimp aquaculture.

## 2. Material and Methods

### 2.1. Isolation of Luminescent V. harveyi Hosts

Water samples collected from *P. vannamei* shrimp hatcheries (*n* = 20) and aquaculture farms (*n* = 12) were screened for the occurrence of luminescent vibrios by spread plating on nutrient agar supplemented with 3% salt [41] and thiosulfate citrate bile sucrose agar (TCBS agar). The plates were incubated at 28 °C for 18 h and observed in the dark to view the luminescence. The luminescent colonies were isolated and purified by streaking on nutrient agar supplemented with 3% salt. The well-isolated colony was re-streaked and observed for luminescence. The luminescent bacteria were assigned to the Vibrio genus based on the results of growth on TCBS agar, Gram’s staining, nitrate reduction, oxidase production, and the Hugh and Leifson glucose oxidation/fermentation test [42] and used as bacterial hosts. The luminescent vibrios were further tested for *V. harveyi* based on biochemical tests, viz., sugar fermentation (arabinose, cellobiose, dulcitol, galactose, glucose, m-inositol, maltose, mannose, raffinose, rhamnose, salicin, sorbitol, and sucrose), amino-acid decarboxylase/dihydrolase (arginine, ornithine, and lysine), salt tolerance (0%, 0.5%, 1%, 3%, 6%, 8%, and 10% NaCl), amylolytic, proteolytic, and lipolytic DNAase activities, and luminescence production [43]

### 2.2. Isolation of Vibriophage

Water samples from *P. vannamei* shrimp hatcheries (*n* = 20) and aquaculture farms (*n* = 12) of Andhra Pradesh, India, and the sewage treatment plant of Visakhapatnam, India, were collected and screened for lytic vibriophages against luminescent vibrios by employing the single-host enrichment method. Briefly, 47.5 mL of water sample was mixed with 12.5 mL of overnight culture of luminescent *V. harveyi*-LV6 as the host strain, which was added to 15 mL of 5×nutrient broth with 3% salt (peptone 5 g L^−1^, beef extract 3 g L^−1^, NaCl 30 g L^−1^) and incubated for 6 h at 28 ± 2 °C. Post-enrichment, the phage-enriched culture was centrifuged (10,000 rpm for 20 min at 4 °C), filtered through a 0.22 µm sterile syringe filter to remove residual bacterial host cells, and the filtrate was tested for vibriophages by spotting 10 µL of the filtrate on NA + 3% salt plates seeded separately with overnight cultures of each of the luminescent vibriohosts. The appearance of clearance at the spotted area indicated the presence of lytic vibriophages.

### 2.3. Purification and Precipitation of Vibriophage

Vibriophage was purified by employing a single-agar method [44], in which 1 mL of phage (filtrate) and 1 mL of luminescent vibrio host (*V. harveyi*-LV6) were mixed and added to 8 mL of molten and cooled soft nutrient agar supplemented with 3% salt (peptone 5 g L^−1^, sodium chloride 30 g L^−1^, and agar-agar 8 L^−1^) and finally poured onto a sterile petri plate. The plates were incubated at 28 °C ± 2 °C for 8 h to obtain plaques. The phage filtrate was serially diluted in SM buffer and analyzed separately. The phage titer is expressed as pfu mL^−1^ and calculated using the following formula:Number of phages (pfu mL^−1^) = Total number of plaques × Dilution factor(1)

The isolated plaques were picked, and the process was repeated three times to obtain purified vibriophage with consistent plaque morphology. Phage precipitation was performed by treating purified vibriophage (25 mL) with polyethylene glycol (10% *w*/*v* PEG 8000 and 1 M NaCl) at 4 °C for 1 h, followed by overnight incubation at −20 °C for 24 h and centrifugation at 10,000 rpm for 20 min at 4 °C [45,46]. The pellet was resuspended in SM buffer (100 mM NaCl, 8 mM MgSO_4_.7H_2_O, 50 mM Tris-Cl, pH 7.5), which constituted the purified and enriched vibriophage, and stored at 4 °C for downstream analysis.

### 2.4. TEM Morphology of Vibriophage-ϕLV6

For morphological analysis, 10 µL of purified and enriched vibriophage (~10^8^ pfu mL^−1^) was loaded on a 200-mesh copper grid, stained with Uranyless 22409, and examined under 15,000× nm magnification at an accelerated voltage of 120 kV using a transmission electron microscope (JEOL Japan) at the National Institute of Animal Biotechnology, Hyderabad, India.

### 2.5. Host Range Determination

The host range of vibriophage-ϕLV6 was determined by performing a spotting assay on 27 luminescent *Vibrio* spp. isolates from *P. vannamei* shrimp hatcheries, viz., LV6, LV20, LV21, LV22, LV23, LV24, LV25, LV26, LV27, LV28, LV29, LV30, LV31, LV32, LV33, LV34, LV35, LV36, LV37, LV38, LV39, LV40, LV41, LV42, LV43, LV44, and LV45.

### 2.6. DNA Extraction and Whole Genome Sequencing of Vibriophage-ϕLV6

Vibriophage-ϕLV6 DNA was extracted and purified with Qiagen’s DNeasy Blood & Tissue Kit [47]. Initially, PEG-precipitated vibriophage (500 µL) was treated with 1.25 µL DNase and RNase (20 mg mL^−1^) and incubated at 37 °C for 1 h; then treated with 1.25 µL proteinase (20 mg mL^−1^) and 25 µL of 10% SDS and incubated at 60 °C for 1 h. The vibriophage-ϕLV6 DNA was extracted as per the kit manufacturer’s instructions and finally suspended in TE buffer. The quality and concentration of the extracted DNA were assessed using the Qubit^®^ dsDNA HS Assay Kit, and the integrity of the DNA was determined by electrophoresis on 1% agarose gel. The whole genome sequencing libraries were prepared using the NEBNext^®^Ultra^TM^ II FS DNA Library Prep Kit for Illumina at ClevergeneBiocorp Private Limited, Bangalore, India. The QC-passed library was diluted to 2 nM and sequenced on the Illumina HiSeq 4000. The high-quality reads were used to assemble the genomes using the HGA genome assembler [48].

### 2.7. Bioinformatic Analysis of Vibriophage-ϕLV6 Genome

The genes in the vibriophage-ϕLV6 genome were predicted and annotated using GeneMarkS, Glimmer, and Prokka [49,50,51]. The functional characteristics of the predicted genes were determined on the NCBI BLASTp platform with the non-redundant protein sequences (nr) database. The presence of transfer RNA (tRNA) was predicted employing Prokka 1.14.6 [50]. The presence of antibiotic-resistance genes was screened in Resfinder 4.1 (https://cge.food.dtu.dk/services/ResFinder/ (accessed on 26 November 2022)) and the presence of bacterial virulent genes was screened in Virulence finder 2.0 (https://cge.food.dtu.dk/services/VirulenceFinder/ (accessed on 26 November 2022)). 

Multiple phage genomes belonging to siphoviruses were compared and visualized using BRIG (Blast Ring Image Generator) with default settings. Comparative genome analysis was performed using ViPTree [52]. The phage genome was categorized into structural modules, DNA metabolism modules, packaging modules, lysis modules, hypothetical modules, and additional functional modules [31,53,54]. Data pertaining to the major capsid protein (*n* = 16) terminase large subunit (*n* = 18) of vibriophages and other phages related to vibriophage-ϕLV6 were downloaded from the NCBI database and aligned using the MUSCLE algorithm. Phylogenetic analysis was performed using MEGA 10.0.5 software [55] based on the major capsid protein and terminase large subunit using the neighbor-joining method with robust 1000 bootstrap replicates. The genome map of vibriophage-ϕLV6 was drawn using Proksee (https://proksee.ca/ (accessed on 2 December 2022)).

### 2.8. In Vitro Determination of Optimum Multiplicity of Infection (MOI) for Determining Phage Infectivity Potential

The multiplicity of infection, i.e., the ratio of the number of vibrophage-ϕLV6 required to lyse luminescent *V. harveyi*, was determined employing the 2-step microtiter plate assay [56]. Briefly, in the 2-step microtiter assay, a broad range of MOIs (ranging from MOI-0.0001 to MOI-10000) were initially tested for their ability to inhibit the growth of target bacteria. A narrow range of effective MOIs from the first step was selected to determine the optimum MOI in the second step. The optimum MOI, i.e., the lowest number of phages required to inhibit the growth of the target bacteria, was determined in the second step. The optimum MOI of vibriophage-ϕLV6 against the luminescent *V. harveyi*-LV6 host was 79 and was previously determined [56]. On similar lines, the optimum MOIs of vibriophage-ϕLV6 against five other luminescent vibrio hosts, viz., LV36, LV38, LV40, LV44, and LV45, isolated from shrimp hatcheries, were determined.

### 2.9. In Vivo Challenge Studies to Assess the Ability of Vibriophage-ϕLV6 to Inhibit the Growth of Luminescent V. harveyi in P. vannamei Shrimp Post-Larvae Tanks

Challenge studies to check the in vivo efficacy of the phage to treat induced luminescent vibriosis were performed in glass tanks containing *P. vannamei* post-larvae. The tank experiments were designed in two variations,
Application of vibriophage-ϕLV6 at optimized MOI against a single luminescent *V. harveyi* host, i.e., LV6;Application of vibriophage-ϕLV6 at optimized MOIs against multiple (*n* = 6) luminescent Vibrio Hosts, i.e., LV6, LV36, LV38, LV40, LV44, and LV45.

#### 2.9.1. Effectiveness of Vibriophage-ϕLV6 Application at Optimized MOI against a Single Luminescent *V. harveyi* Host

Glass tanks filled with 27 ppt seawater (10 L) and *P. vannamei* post-larvae of PL-11 size (*n* = 250) per tank were used for the in vivo study. Tank-1 (control) contained only shrimp post-larvae; Tank-2 (bacteria control) was spiked with luminescent *V. harveyi*-LV6 at an 8.6 × 10^6^ cfu mL^−1^ concentration. Tank-3 (phage control) was inoculated with vibriophage-ϕLV6 (8.0 × 10^6^ pfu mL^−1^). Tank-4 (treatment tank) was spiked with luminescent *V. harveyi*-LV6 and simultaneously treated with vibriophage-ϕLV6 at an MOI of 80 (80 pfu phage to 1 cfu bacteria). All the tanks were kept at ambient temperature (28–30 °C) under illuminated conditions with continuous aeration. Feed was not provided to the post-larvae during the experiment period. Water samples were taken at hourly intervals for 6 h and checked for OD_600_ values; the total vibrio counts and phage activity were checked in the water after 4 h, and shrimp post-larval survivability was determined after 24 h of exposure. 

#### 2.9.2. Effect of Vibriophage-ϕLV6 on the Growth of Multiple Luminescent Vibrio Hosts (*n* = 6)

An in vivo experiment was conducted in glass tanks containing 25 ppt salinity sea water (1 L) and *P. vannamei* post-larvae of PL-3 size (*n* = 100) per tank. The bacteria control tanks were spiked with six luminescent *Vibrio* spp. isolates (LV36, LV38, LV40, LV44, LV45, and LV6; 10^9^ cfu mL^−1^). The vibriophage-treatment tanks were spiked with six luminescent vibrio bacteria at a concentration of 10^9^ cfu mL^−1^ (each bacteria) and simultaneously treated with vibriophage-ϕLV6 at pre-determined MOIs (MOI-79 for the LV6 host, MOI-41.5 for the LV40 host, MOI-33.6 for the LV36 host, MOI-29.3 for the LV38 host, MOI-1.5 for the LV45 host, and MOI-0.7 for the LV44 host). The control tanks were spiked with neither bacteria nor bacteriophage. All the tanks were kept at ambient temperature (28–30 °C) under illuminated conditions with continuous aeration. Feed was not provided to the post-larvae during the experiment period. The total vibrio counts and post-larvae survivability were checked for 48 h.

### 2.10. Lytic Ability of Vibriophage-ϕLV6 under Different Salinity Conditions

In the shrimp aquaculture system of India, the salinity of the water in the shrimp hatcheries is maintained between 25 and 35 ppt (2.5 and 3.5%) but the farming of *P. vannamei* shrimp in aquaculture farms is carried out at different salinities ranging from 4 ppt to 45 ppt [57]. In order to check the applicability of the vibriophage-ϕLV6, both in hatcheries and different farming conditions of *P. vannamei*, a salt gradient experiment on determining the lytic activity of vibriophage was taken up. Tubes containing nutrient broth with different salt concentrations, viz., 0.5%, 1%, 2%, 3%, 4%, and 5%, were inoculated with vibriophage-ϕLV6 at a 10% level and incubated at 28 ± 2 °C. The tubes were taken out every 3 days for 30 days and checked for the lytic activity of vibriophage-ϕLV6 by the spotting method.

### 2.11. Vibriophage-ϕLV6 Activity against Luminescent V. harveyi Host LV6 at Different Salt Gradients

Nutrient broth with different salt concentrations, viz., 0%, 0.25%, 0.5%, 1%, 1.5%, 2%, 2.5%, and 3%, was prepared, and 240 µL of NB with a specific salt concentration was loaded in triplicate wells of a 96-well microtiter plate reader (BioTek, Winooski, VT, USA). Vibriophage-ϕLV6 (30 µL) and *V. harveyi* host (30 µL) were inoculated at a pre-determined MOI of 80 into each well of a 96-well microtiter plate.

Three controls, viz., bacterial control (without phage), phage control (without bacteria), and media control (without bacteria and phage), were also introduced in triplicate wells at the same salt gradients. The OD_600_ readings were taken at 30 min intervals for 4 h. 

### 2.12. Storage Stability of Vibriophage

The vibriophage-ϕLV6 suspended in SM buffer was stored at 4 °C for one year. Samples were drawn intermittently, and the counts (pfu mL^−1^) of vibriophage-ϕLV6 were determined by performing the single-agar method [44].

### 2.13. Statistical Analysis

The results of post-larval survivability and vibrio counts in phage-treatment tanks and bacterial-challenged tanks were analyzed statistically employing an unpaired *t*-test and a chi-square test to test the difference at a 5% level of significance using InVivoStat, Version 4.7 [58]. 

## 3. Results and Discussion

### 3.1. Isolation of Luminescent Vibrios

A total of 27 luminescent vibrios were isolated from *P. vannamei* shrimp hatchery water samples that showed characteristic luminescence activity. Luminescent vibrios were not detected in water samples from shrimp aquaculture farms. One luminescent vibrio (LV6) that was isolated from a luminescent vibriosis-infected *P.s vannamei* post-larvae tank was identified as *V. harveyi* based on the results of biochemical and morphintorial analysis. The remaining luminescent vibrios were identified to the genus level and categorized as luminescent *Vibrio* spp. (Figure 1a,b).

### 3.2. Isolation and Lytic Spectrum of Vibriophage

A lytic phage named vibriophage-ϕLV6 was isolated from the water of the post-larvae tank of a luminescent vibriosis-infected *P. vannamei* shrimp hatchery. Vibriophage-ϕLV6 showed a clear lytic zone against luminescent *V. harveyi*-LV6 (Figure 2) after single-host enrichment and yielded pinpointed plaques with a diameter of <1 mm on the single-agar method. The vibriophage count after purification, PEG precipitation, and suspension in SM buffer was 3.42 × 10^10^ pfu mL^−1^. PEG precipitation increased the concentration of the vibriophage 100 times. None of the remaining 19 water samples collected from different shrimp hatchery sites, 20 water samples from shrimp aquaculture farms, and one water sample from a sewage treatment plant revealed the presence of vibriophage against luminescent *V. harveyi*. The vibriophage-ϕLV6 showed lytic activity against one luminescent *V. harveyi* (LV6) and five isolates of luminescent *Vibrio* spp. from shrimp hatcheries, viz., LV36, LV38, LV40, LV44, and LV45. 

### 3.3. TEM Morphology of Vibriophage-ϕLV6

The morphology of vibriophage-ϕLV6 observed under transmission electron microscopy (Figure 3) revealed that the phage had an icosahedral head (~73 nm) and a long filamentous non-contractile tail (~191 nm), respectively, suggesting that the vibriophage-ϕLV6 might belong to siphoviruses. 

### 3.4. Vibriophage-ϕLV6 Genome Characterization and Phylogenetic Analysis

The genome size of the vibriophage-ϕLV6 was 79,862 bp with a G+C content of 48%. The coding sequences (CDS) accounted for 93% of the vibriophage genome (74,178 bp) and coded for proteins with known functions (41%), and hypothetical proteins (52%). A total of 107 putative ORFs were predicted in the genome of vibriophage-ϕLV6 with gene lengths ranging between 129 and 4146 bases. The details of the ORFs and the predicted proteins are presented in Figure 4 and Table 1. Out of 107 ORFs, 31 of the ORFs in the vibriophage-ϕLV6 genome had predicted functions, and 75 ORFs had currently unknown functions, i.e., hypothetical proteins. One tRNA coding for lysine was found in the genome of vibriophage-ϕLV6. No antibiotic resistance genes or bacterial virulence genes were found in the genome of vibriophage-ϕLV6. 

### 3.5. Intergenomic Comparison of Vibriophage-ϕLV6

#### 3.5.1. Phylogenetic Analysis

Phylogenetic trees were constructed targeting single gene analysis of major capsid protein (Figure 5a) and terminase large subunit protein (Figure 5b). Results from the two phylogenetic trees indicated that vibriophage-ϕLV6 is closely related to vibriophage-V-YDF132, vibriophage-VH2_2019, and vibriophage-vB_VpS_PG28.

#### 3.5.2. Multiple Genome Alignments

Vibriophage-ϕLv6 (NCBI Accession ID: OP918262.1) had genome sequence similarity of 79.89%, 74.97%, and 74.44% to vibriophage-vB_VpS_PG28 (NCBI Accession ID: MT735630.2), vibriophage-V-YDF132 (NCBI Accession ID: 0N075462.1) and vibriophage-VH2_2019 (NCBI Accession ID: MN79428.1). The Bacterial and Archaeal Subcommittee (BAVS) considers that any genus having similarity greater than 50% can be considered a coalescent group [59]. Multiple sequence alignment was performed with Vibriophage-ϕLV6 and its closely related vibriophages, i.e., vibriophage-V-YDF132, vibriophage-VH2_2019, and vibriophage-vB_VpS_PG28 (Figure 6). 

#### 3.5.3. Whole Genome Comparison

The whole genome sequences of the vibriophages used in phylogenetic tree construction of terminase subunit and major capsid protein were obtained from NCBI and compared with the genome of vibriophage-ϕLV6. The results were similar to the phylogenetic analysis, i.e., vibriophage-V-YDF132 (isolated from a group fish farm in China and active against *V. harveyi*), vibriophage-VH2_2019 (isolated from Hatches Creek, USA, and active against *V. natriegens*), and vibriophage-vB_VpS_PG28 (isolated from sewage at a seafood market in China and active against *V. parahaemolyticus*) shared more similarity with vibriophage-ϕLV6 compared to other phages (Figure 7). 

A whole genome homologs similarity search revealed that vibriophage-vB_VpS_PG28 infecting *V. parahaemolyticus* shared the highest similarity to vibriophage-ϕLV6, but with a similarity coverage of only 79.89%, followed by two other vibriophages, i.e., vibriophage-V-YDF132 and vibriophage-VH2_2019, with sequence similarities of 74.97% and 74.44%, respectively. Vibriophage-ϕLV6 harbored replication, regulation, structural, and packaging modules similar to those of vibriophage-vB_VpS_PG28 and vibriophage-VH2_2019 but lacked a lysis module. In contrast, vibriophage-ϕLV6 and vibriophage-V-YDF132 shared all the modules (Figure 8). Vibriophage-ϕLV6 had one tRNA encoding for lysine.

#### 3.5.4. Comparison of the Genome of Vibriophage-ϕLV6 with the Genomes of Other Vibriophages Active against *V. harveyi* Clade

The whole genome features of vibriophage-ϕLV6 were compared to the features of other vibriophages that were previously reported to be active on *V. harveyi* clade bacteria, viz., siphophage-VHS1, vibriophage-vB_VhaM_pir03, vibriophage-VB_VhaS_PcB-1G, vibriophage-VB_VcaS_HC, and vibriophage-Virtus. The genome size of vibriophage-ϕLV6 (79.86 kb) was similar to that of siphophage-VHS1 (81.5 kb), Vibriophage-VB_VcaS_HC (81.6 kb), and vibriophage-virtus (82.96 kb), but was smaller than the jumbo vibriophage-vB_VhaM_pir03 (286.3 kb) (Table 2). The G+C content of vibriophage-ϕLV6 (48%) was comparable to that of the previously reported vibriophages (43.6% to 47.6%). The ORFs with predicted function were relatively higher in vibriophage-ϕLV6 (138) compared to other vibriophages with similar genome sizes (121–127 ORFs). The jumbo-sized vibriophage-vB_VhaM_pir03, with a genome size larger than 200 kb, had 137 ORFs. Moreover, tRNA was reported only in vibriophage-ϕLV6 that encoded for lysine.

The vibriophage-ϕLV6 genome was negative for antimicrobial-resistance determinants (ARGs), integrase, and bacterial virulence genes, making it a suitable candidate for in vivo phage applications to control luminescent vibriosis in shrimp aquaculture. 

### 3.6. Vibriophage-ϕLV6 Proteome

Vibriophage-ϕLV6 has six ORFs, i.e., ORF 3, ORF 6, ORF 7, ORF 8, ORF 9, and ORF 11, predicted for functions related to the structural composition of the phage. Vibriophage-ϕLV6 has dedicated machinery for 7 ORFs, i.e., ORF 62, ORF 68, ORF 72, ORF 86, ORF 92, ORF 94, and ORF 96, involved in phage DNA metabolism. ORF 82, which encodes for the terminase large subunit, plays a vital role in viral genome packaging. The terminase large subunit has two subunits: a smaller subunit involved in viral DNA packaging and a larger subunit involved in ATPase and endonuclease activities. 

### 3.7. Determination of Optimum Multiplicity of Infection

The optimization of MOI is important to determine the lowest number of phages required to inhibit the growth of a specific bacteria. The luminescent *V. harveyi* isolates that were susceptible to vibriophage-ϕLV6 in the spot assay were selected for MOI determination. The optimum MOI for the isolate LV6 was previously reported by us at MOI-79 [56]. Similarly, the optimum MOI of vibriophage-ϕLV6 to inhibit the growth of the remaining five susceptible luminescent *Vibrio* spp. isolates, viz., LV36, LV38, LV40, LV44, and LV45, was determined using the two-step microtiter plate assay. In the two-step microtiter assay, a narrow range of MOIs were selected in the first step, and the optimum MOI was determined in the second step (Figure 9a,b). The narrow range of MOIs out of the nine MOIs (0.0001 to 10,000) that were selected in the first step ranged between 6.725 and 672.5 for LV36; 5.854 and 585.4 for LV38; 8.3 and 83 for LV40; 0.01 to 1.42 for LV44; and 0.03 to 3 for LV45. The optimum MOIs of vibriophage-ϕLV6 obtained in the second step were 41.5 for LV40, 33.6 for LV36, 29.3 for LV38, 1.5 for LV45, and 0.7 for LV44. These optimized MOIs were applied for challenge studies in glass tanks. 

### 3.8. Challenge Studies to Test the In Vivo Lytic Ability of Vibriophage-ϕLV6

Vibriophage-ϕLV6 was employed at an optimized MOI to control the growth of luminescent *V. harveyi* and luminescent *Vibrio* spp. in tanks containing post-larvae of *P. vannamei* shrimp that were spiked with either single or multiple isolates of luminescent *Vibrio* spp. 

#### 3.8.1. Effectiveness of Vibriophage-ϕLV6 Treatment at an Optimized MOI of 80 against a Single Luminescent *V. harveyi*-LV6

There was a continuous increase in OD_600_ in the bacteria-control tank, indicating uncontrolled proliferation of bacteria, whereas the vibriophage-treated tanks showed a negligible increase in OD_600_ until 6 h of exposure. The luminescent bacteria count was very high in the bacterial-control tank (1.02 × 10^8^ cfu mL^−1^) after 4 h of exposure; however, the luminescent bacteria count was less than 300 cfu mL^−1^ in vibriophage-ϕLV6 treated tanks (Figure 10). The vibriophage count ranged between 3.6 × 10^9^ and 3.7 × 10^9^ pfu mL^−1^ in phage-treated tanks, whereas no vibriophage was detected in bacteria-control tanks. There was a continuous increase in the OD_600_ values of the water in the bacteria-control tank, indicating uncontrolled proliferation of bacteria. However, the vibriophage-ϕLV6 treated tank showed a negligible increase in OD_600_ values, lower luminescent bacterial counts, but very high counts of vibriophage and higher post-larvae survival. The results of the tank (10 L) indicated the effectiveness of employing the vibriophage-ϕLV6 in controlling the growth of luminescent *V. harveyi*, and the optimum MOI determined by the two-step microtiter plate method was sufficient to control the growth for 6 h.

#### 3.8.2. Effect of Vibriophage-ϕLV6 on Inhibiting the Growth of Multiple Luminescent Vibrio Hosts (*n* = 6)

The shrimp post-larvae mortality was higher in bacteria-challenged tanks (37.5% ± 3%) compared to phage-treated tanks (9.5% ± 3%) and control tanks (8% ± 1%). In other words, significantly higher survivability of the shrimp post-larvae was observed in the phage-treated tank (Figure 11) compared to the bacteria-challenged group (unpaired *t*-test and chi-square test, *p* < 0.05). The sucrose non-fermenting vibrio loads were distinctly higher in bacteria-spiked tanks (357,100 cfu mL^−1^) compared to vibriophage-treated tanks (1000 cfu mL^−1^). Phage activity was detected only in vibriophage-treated tanks but not in control or bacteria-spiked tanks. The results indicate that Vibriophage-ϕLV6 effectively reduced the numbers of multiple luminescent vibrios and reduced the mortality of shrimp post-larvae. 

### 3.9. Survivability of Vibriophage under Different Salinity Conditions

The vibriophage-ϕLV6 survived under different salt conditions of 5 ppt, 10 ppt, 20 ppt, 30 ppt, 40 ppt, and 50 ppt, indicating their applicability in brackish water and marine waters. Vibriophage-ϕLV6 survived and exhibited its lytic activity for 30 days (maximum period tested) at both 28 °C and 35 °C.

### 3.10. Stability Testing of Vibriophage-ϕLV6 Activity against Luminescent V. harveyi Host LV6 at Different Salt Gradients

*V. harveyi*-LV6 (bacteria controls) did not show any growth at 0%, very weak growth at 0.25% salt concentration, and relatively weak growth at 0.5% salt concentration. The growth of *V. harveyi*-LV6 was optimal at salt concentrations between 1% and 3%, yielding an OD_600_ value of ~0.3 to 0.4 (Figure 12). This growth pattern vis-à-vis salt concentration was on expected lines, as *V. harveyi* is a halophilic organism and salt is integral to its growth [30]. It was pertinent to note that the lytic activity of vibriophage-ϕLV6 when applied at an MOI of 80 against *V. harveyi*-LV6 was not affected by the different salt concentrations (0% to 3%) and efficiently halted the growth of *V. harveyi*-LV6 as evidenced by lower (~0.1) OD_600_ that were similar to the media controls and phage controls at the end of 240 min of incubation (Figure 12). 

### 3.11. Storage Stability of Vibriophage-ϕLV6

Concentrated suspension of vibriophage-ϕLV6 stored at 4 °C did not show any reduction in phage numbers for 9 months of storage as the plaque counts obtained on single agar remained almost similar (10^10^ pfu mL^−1^), but a slight reduction in the phage numbers (less than one log) was observed at the end of 12 months of storage at 4 °C (Table 3). 

## 4. Discussion

The use of bacteriophages as therapeutic agents in aquatic animal-health management has gained renewed interest due to the emergence of resistance in pathogenic bacteria towards antibiotics and safety issues related to antibiotic residues in food products. Globally, shrimp farming is increasingly contributing to animal protein requirements, but farm productivity is adversely affected by diseases caused by bacteria of the genus Vibrio. *V. harveyi* is the major causative agent of luminescent vibriosis in shrimp hatcheries and aquaculture farms. In the present study, a vibriophage, named vibriophage-ϕLV6, was isolated from the water of a *P. vannamei* shrimp hatchery. Vibriophage-ϕLV6 showed in vitro lytic activity against luminescent *V. harveyi* (LV6) that was isolated from a shrimp hatchery affected with luminescent vibriosis. Vibriophage-ϕLV6 produced pinpoint plaques on soft-agar plates seeded with the bacterial host. Misol et al. (2020) also reported that vibriophage-vB-VhaM-pir03 produced pin-hole plaques with a diameter of 0.27 ± 0.05 mm on *V. harveyi*-seeded plates. A 100-fold increase in the concentration of vibriophage-ϕLV6 was achieved by PEG precipitation, and a similar increase in the concentration of coliphages by PEG precipitation was reported [46]. The host spectrum of vibriophage-ϕLV6 (six luminescent Vibrios) was relatively lower compared to recently reported *V. harveyi* bacteriophages. Vibriophage-Virtus, isolated from the water of a fish brood stock section in Crete, Greece, could infect 8 of the 16 strains of *V. harveyi* [39], and vibriophage-B_VhaM_pir03, isolated from the water of the Port of Piraeus, Greece, showed lytic activity against 31 AMR strains of *V. harveyi*, *V. alginolyticus*, *V. campbellii*, and *V. owensii* [38]. Most of the phages that were isolated against *V. harveyi* are reported to be lytic, but two bacteriophages, viz., VHML [4] and VHS1 [60], were found to be temperate.

Transmission electron microscopy images indicate that vibriophage-ϕLV6 belongs to the morphological group of siphoviruses. The bacterial viruses subcommittee of the International Committee on Taxonomy of Viruses (ICTV) has recently abolished the morphology-based families of bacteriophages, viz., *Myoviridae*, *Podoviridae*, and *Siphoviridae*, due to their polyphyletic nature and non-reflection of shared evolutionary histories. All the tailed phages with icosahedral capsids and dsDNA genomes are now grouped under the class *Caudoviricetes* [61,62,63,64,65]. The terms myovirus, podovirus, and siphovirus can be used to represent distinctive morphological features and retain their historical reference. The process of assigning myoviruses, podoviruses, and siphoviruses into genomically coherent families has been initiated [65]. Though vibriophage-ϕLV6 reveals a siphovirus morphology, it is grouped under *Caudoviricetes* as per the new ICTV classification. Most of the phages that showed lytic activity against *V. harveyi* were reported to be siphoviruses [29,32,34,39,66,67,68], followed by myoviruses [34,38,68,69,70].

The genome analysis of vibriophage-ϕLV6 indicated a genome size of 79.8 kb with a G+C content of 48% that was comparable to the previously reported vibriophages (43.6% to 47.6%). The genome of vibriophage-ϕLV6 was highly functional. The ORFs with predicted function were relatively higher in vibriophage-ϕLV6 (138) compared to other vibriophages with similar genome sizes (121–127 ORFs). Even the jumbo-sized vibriophage-vB_VhaM_pir03, with a genome size larger than 200 kb, had only 137 ORFs. Vibriophage-ϕLV6 harbored replication, regulation, structural, and packaging modules. Though the vibriophage-ϕLV6 genome did not reveal a lysis module, it carried a sufficient number of genes that encode for early DNA metabolism, which play an essential role in early viral infection similar to vibriophage-V-YDF132 [39]. Vibriophage-ϕLV6 possesses an ORF7 that encodes for a structural protein (tail tube measure protein). Wu et al. (2020) stated that tail tubular protein, encoded by vibriophage-PcB-1G, plays a critical role in bacterial-cell lysis [32], suggesting that Vibriophage-ϕLV6 can mediate bacterial-cell lysis through the tail tubular system. Vibriophage-ϕLV6 possessed a single tRNA that encoded lysine. Many vibriophages, such as *Vibrio parahaemolyticus* phage-seahorse and KVP-40 carried high numbers of tRNAs, which may provide the phage with a small degree of autonomy when it comes to the translation of its own genes [71]. It is pertinent to note that more than half of the 107 ORFs of vibriophage-ϕLV6 code for hypothetical proteins whose function is currently unknown. Research efforts are needed to decipher the true function of these hypothetical proteins, as the viral genome machinery is relatively small and has no reason to burden itself with unwanted proteins.

Phylogenetic trees constructed with conserved proteins in the bacteriophage genomes, viz., the major capsid protein and the terminase large subunit protein [72], indicated that vibriophage-ϕLV6 is closely related to vibriophage-V-YDF132, vibriophage-VH2_2019, and vibriophage-vB_VpS_PG28, asserting that their origin is from a common ancestor. Multiple genome alignments showed that vibriophage-ϕLv6 had genome sequence similarity of ~75% to vibriophage-vB_VpS_PG28, vibriophage-V-YDF132, and vibriophage-VH2_2019. The proteome of vibriophage-ϕLV6 had a dedicated machinery of ORFs involved in phage DNA metabolism and viral genome packaging. The proteome of vibriophage-ϕLV6 is similar to the proteome of vibriophage-V-YDF132 [31]. Vibriophage-ϕLV6 possessed an auxiliary metabolic gene that encodes for pyruvate phosphate dikinase (PPDK), an essential component of the Embden–Meyerhof–Parnas (EMP) glycolytic pathway that was also reported in the vibriophages belonging to siphovirus [69,73]. Genomes of the marine vibriophages isolated from nutrient-deficient environments were abundant in auxiliary metabolic genes compared to those isolated from nutrient-rich environments [59]. 

Horizontal gene transfer (HGT) occurs between phages and bacterial populations through either generalized or specialized transductions [70,74]. Prior to their therapeutic application, the profiling of vibriophages for genomic traits is an essential pre-requisite to ward off an unwanted increase in the virulence of their hosts [16,59,70]. Vibriophage-ϕLV6 appears to be a suitable candidate phage for in vivo phage applications to control luminescent vibriosis as it does not carry antimicrobial resistance determinants or bacterial virulence genes. Moreover, it does not have an integrase gene associated with phage lysogeny. The vibriophage-ϕLV6 produces clear plaques against the host bacteria and reduced the counts of host bacteria in microtiter plate assays, and the reduction in growth was proportional to the number of phages used (i.e., lower bacterial growth at higher MOIs). These results indicate the vibriophage-LV6 was lytic and not lysogenic. Siphophage-VHS was reported to carry a shrimp haemocyte agglutination gene [55]. Siphophage-VHS1 and vibriophage-ϕLV6 were tested on *V. harveyi* in Pacific white shrimp (*P. vannamei*) tanks, while vibriophage-vB_VhaM_pir03 was tested in brine shrimp (Artemia) culture. On the other hand, vibriophage-VB_VhaS_PcB-1G and vibriophage-virtus were tested on *V. harveyi* in finfish tanks.

Several studies have demonstrated the effectiveness of vibriophages in treating vibriosis in a variety of animal models [17,29,38,39,75,76,77,78,79]. Karunasagar et al. (2007) reported that vibriophages resulted in higher survival rates (80%) of black tiger shrimp (*P. monodon*) in hatcheries compared to survival achieved by conventional antibiotic treatment (40%). vB_VhaM_pir03 when applied to *Artemia naupli*, increased the survival rates of larvae in the phage-treated group to 15–20% in 48 h than the *V. harveyi* bacterial control group [38]. Droubogiannis, and Katharios, reported 35% survival of gilthead seabream larvae in a single dose of phage application, while Vinod et al. (2006) reported higher survival of *P. monodon* in double dose (80%) phage treated groups compared to the *V. harveyi* challenged control group. Misol et al. (2020) observed that during vibriophage treatment, the bacterial population infected at MOI-10 showed the lowest growth. Droubogiannis and Raveearios (2022) reported that the growth of bacteria was inhibited within 2 h of post-infection with vibriophage at an MOI of 100 but took longer time at lower MOIs of 0.1, 1, and 10. Here we report that the application of vibriophage-ϕLV6 to post-larvae of *P. vannamei* challenged with *V. harveyi* at an optimized MOI of 80 resulted in a steep decrease in the luminescent *V. harveyi* counts in phage-treated tanks compared to counts in bacterial-challenged tanks. Shrimp post-larvae survivability was higher in phage-treated tanks compared to bacteria-spiked tanks. As vibriophage-ϕLV6 phage suspension resulted in higher post-larval survival, decreased luminescent vibrio loads, and decreased sucrose non-fermenting vibrio counts, it can be considered therapeutic to control luminescent vibriosis in hatcheries and aquaculture systems. However, employing a cocktail of phages can overcome possible phage resistance and simultaneously inhibit several strains of luminescent vibrios. The survivability of vibriophage-ϕLV6 under different salt conditions (0.5% to 5%) indicates its applicability in shrimp hatcheries and growth ponds.

## 5. Conclusions

The present study demonstrates the isolation and genomic characterization of vibriophage-ϕLV6 and assesses its in vitro and in vivo lytic ability against luminescent *Vibrio harveyi*. There is a paucity of complete genome data on vibriophages against *V. harveyi* available in the NCBI database, and in this context, the genomic information of vibriophage-ϕLV6 adds new information from India. The vibriophage-ϕLV6 genome codes for many hypothetical proteins, and research efforts are needed to elucidate their function for a complete understanding of the vibriophage. In vitro and in vivo inhibition trials with vibriophage-ϕLV6, indicated a decrease in luminescent vibrio loads and higher shrimp post-larval survival in phage-treated tanks compared to bacteria-control tanks, suggesting that vibriophage-ϕLV6 can be a potential alternative to antibiotics in reducing luminescent vibriosis in shrimp aquaculture. Prior to phage therapy becoming a common practice for aquatic animal health management in aquaculture, issues such as mass production of bacteriophages, the designing of phage cocktails for warding off phage resistance, the creation of phage repositories, etc., must be addressed.

## Figures and Tables

**Figure 1 viruses-15-00868-f001:**
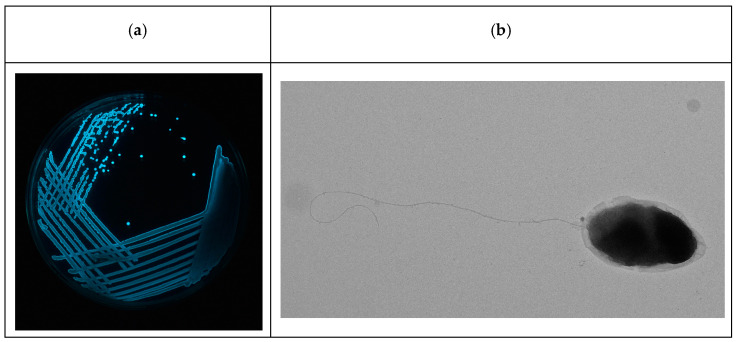
Luminescent V. harveyi-LV6 isolated from a luminescent vibriosis infected *P. vannamei* shrimp hatchery. (**a**) Luminescent colonies on nutrient agar with 3% salt, observed under complete darkness. (**b**) Transmission electron microscopic (TEM) image of *V. harveyi*-LV6 (6000× magnification).

**Figure 2 viruses-15-00868-f002:**
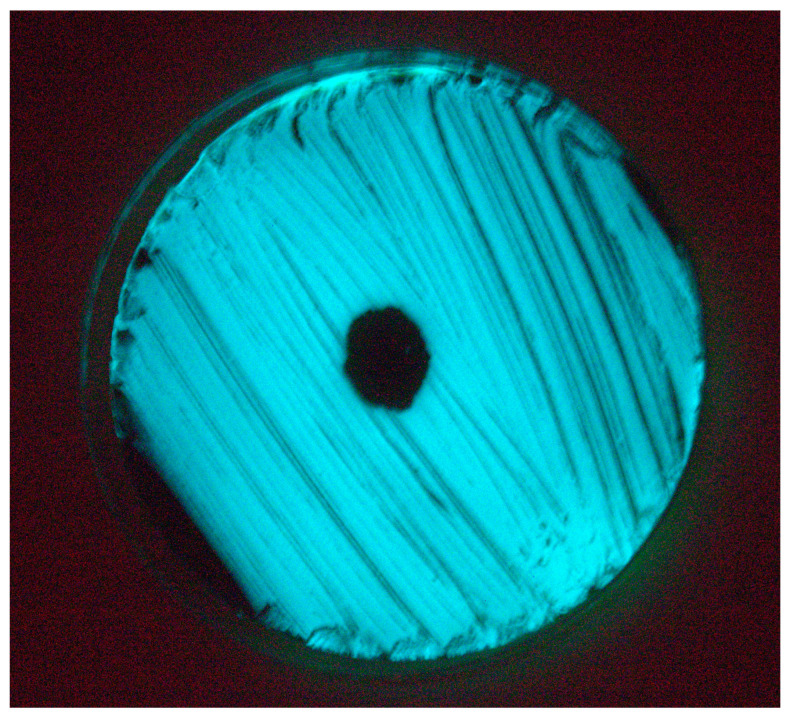
Lytic activity of vibriophage-ϕLV6. Clear lytic zone on the lawn of luminescent *V. harveyi* in spotting assay observed under darkness.

**Figure 3 viruses-15-00868-f003:**
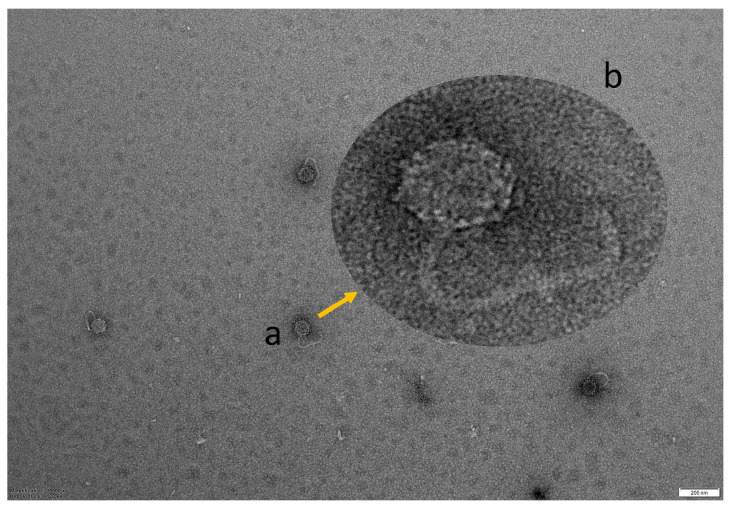
Transmission electron microscopy image of vibriophage-ϕLV6. The phage shows an icosahedral head and a long flexible tail. (**a**) TEM image of vibriophage at 15,000× magnification (scale 200 nm). (**b**) Enlarged image of vibriophage-ϕLV6.

**Figure 4 viruses-15-00868-f004:**
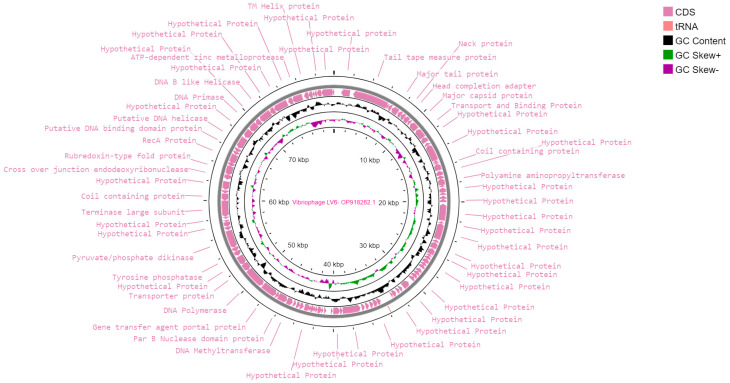
Genome map of Vibriophage-ϕLV6.

**Figure 5 viruses-15-00868-f005:**
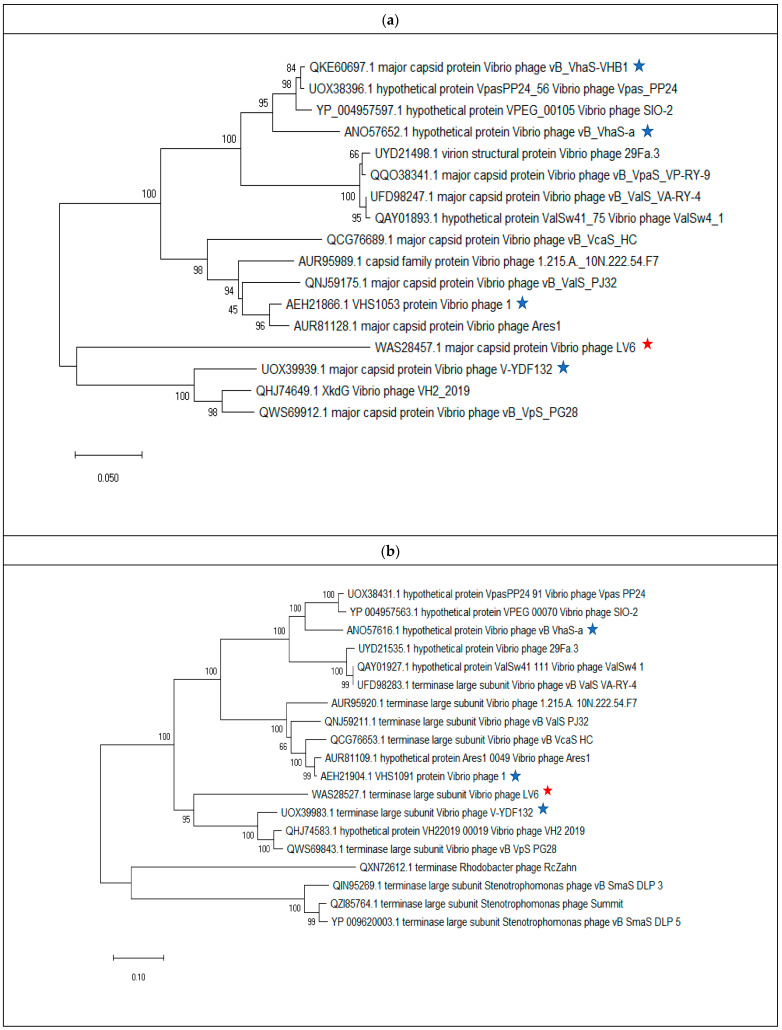
Phylogenetic trees of relatedness of vibriophage-ϕLV6 with other similar Vibriophages belonging to *Siphovirus* and *Caudoviricetes*. (**a**) Analysis based on major capsid protein. (**b**) Analysis based on terminase large subunit. The red pentagram indicates Vibriophage-ϕLV6. The blue pentagram indicates other *V. harveyi* phages.

**Figure 6 viruses-15-00868-f006:**
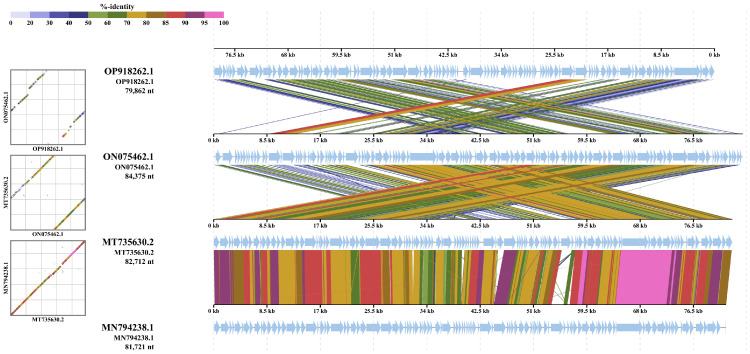
Multiple genome alignment of Vibriophages. The whole genomes of vibriophage-ϕLV6 (OP918262.1), vibriophage-V-YDF132(ON075462.1), vibriophage-VH2_2019 (MT735630.2) and vibriophage-vB_VpS_PG28 (MN794238.1) compared using ViP tree.

**Figure 7 viruses-15-00868-f007:**
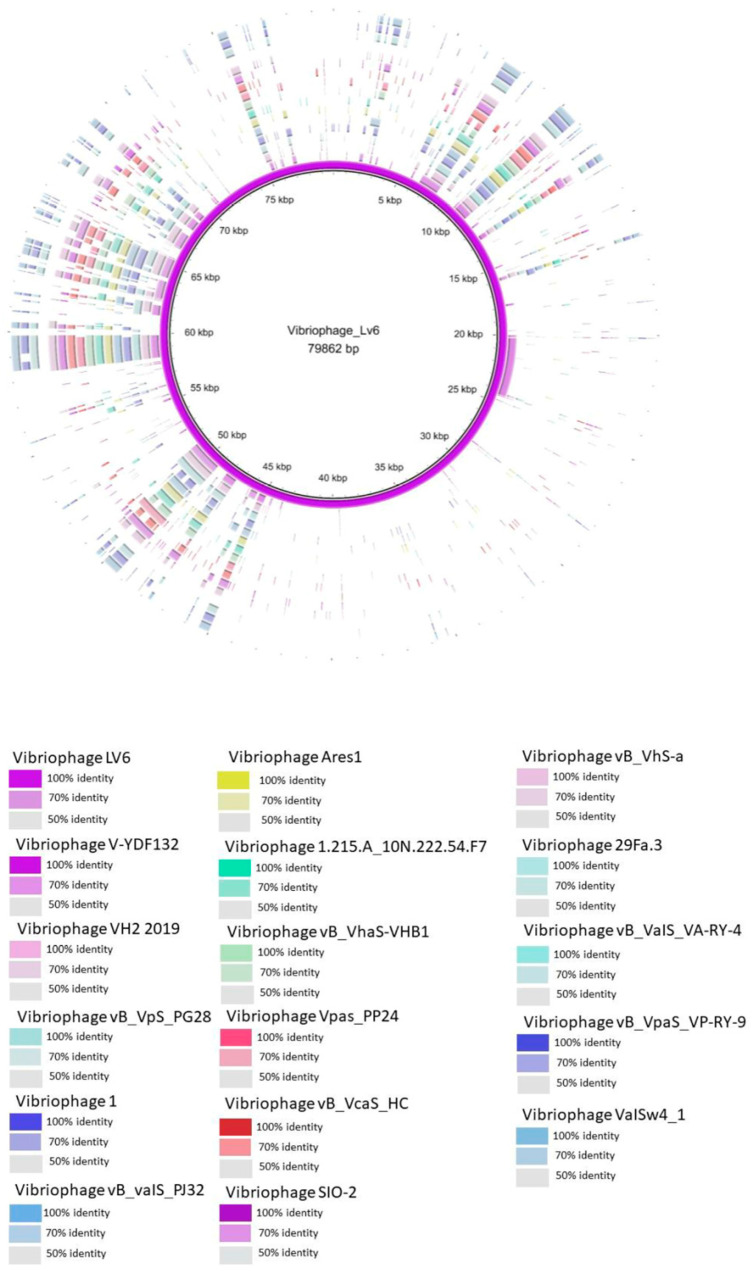
Circular genome comparison of Vibriophage-ϕLV6 against 16 other related vibriophages. The inner first ring in purple color represents the genome of Vibriophage-ϕLV6 and the fragments surrounding the purple ring in different colors represents the similarity sequences shared with vibriophage-ϕLV6 and other 16 vibriophage genomes data obtained from NCBI.

**Figure 8 viruses-15-00868-f008:**
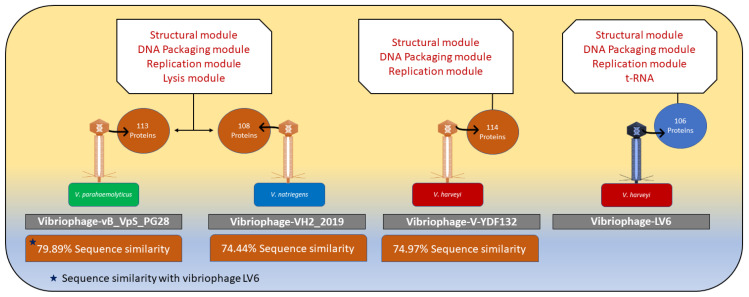
Comparison of lytic machinery among the related Vibriophages.

**Figure 9 viruses-15-00868-f009:**
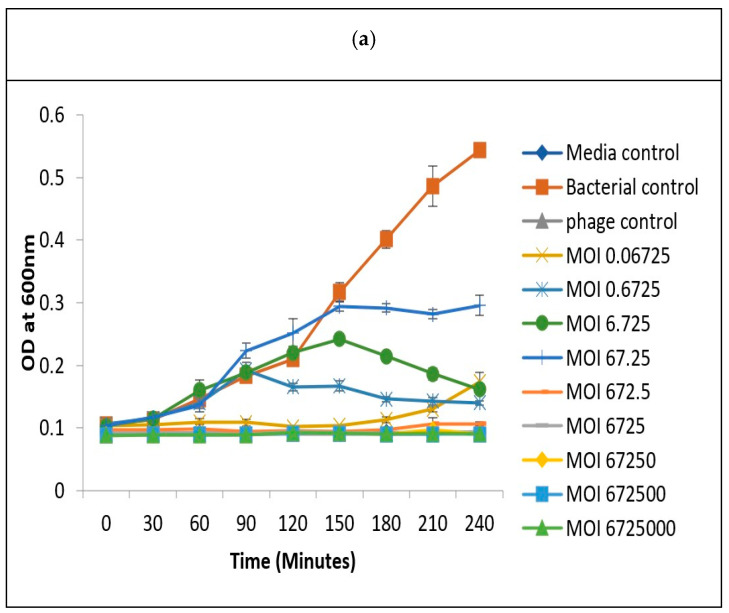
Optimum MOI determination of Vibriophage-ϕLV6 against luminescent *Vibrio* spp.-LV36 in the two-step microtiter plate assay. (**a**) broad range MOIs (step 1). (**b**) narrow range MOIs (step 2).

**Figure 10 viruses-15-00868-f010:**
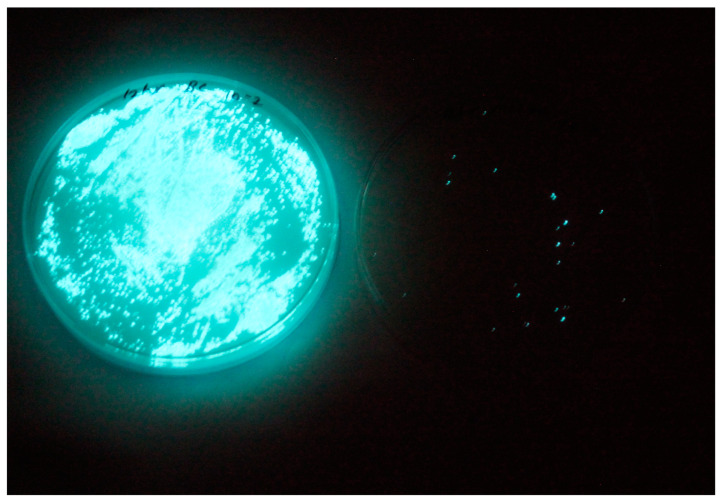
Vibriophage-ϕLV6 treatment at an optimized MOI of 80 against a single luminescent *V. harveyi*-LV6. Very high loads of luminescent bacteria in the water of bacteria-control tank (Petri plate 1) compared to water in vibriophage-treated tanks (Petri plate 2).

**Figure 11 viruses-15-00868-f011:**
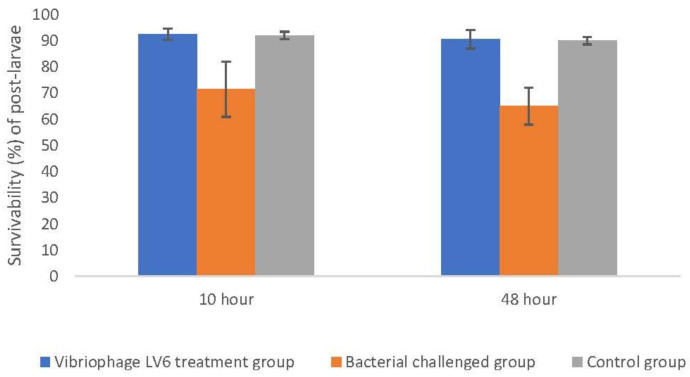
Survival of *P. vannamei* post-larvae in vibriophage-ϕLV6 treated tanks.

**Figure 12 viruses-15-00868-f012:**
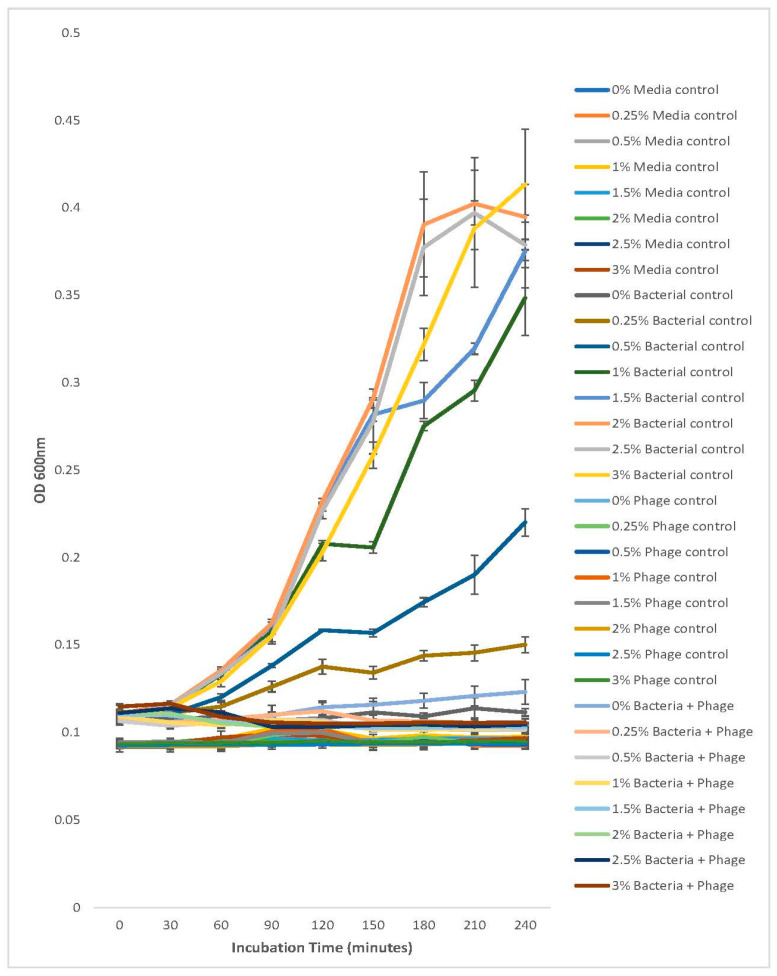
Stability testing of vibriophage-ϕLV6 activity against luminescent *V. harveyi* host-LV6 at different salt gradients.

**Table 1 viruses-15-00868-t001:** ORFs of Vibriophage-ϕLV6 and its predicted proteins including gene length.

Sl.No	ORF/tRNA	Start Base	End Base	Nucleotide Length	Predicted Protein	Category
1	ORF 1	847	1851	1005	Hypothetical Protein	Currently unknown
2	ORF 2	1864	2250	387	Hypothetical Protein	Currently unknown
3	ORF 3	2247	6392	4146	Tail tape measure protein	Structural
4	ORF 4	6405	6554	150	Hypothetical Protein	Currently unknown
5	ORF 5	6605	7009	405	Hypothetical Protein	Currently unknown
6	ORF 6	7086	7883	798	Major tail protein	Structural
7	ORF 7	7941	8390	450	Tail completion protein	Structural
8	ORF 8	8387	8857	471	Neck protein	Structural
9	ORF 9	8857	9462	606	Head completion adapter	Structural
10	ORF 10	9480	9794	315	Hypothetical Protein	Currently unknown
11	ORF 11	9868	10,818	951	Major capsid protein	Structural
12	ORF 12	10,886	12,040	1155	Transport and Binding Protein	Additional function
13	ORF 13	12,041	12,667	627	Hypothetical Protein	Currently unknown
14	ORF 14	12,789	13,115	327	Hypothetical Protein	Currently unknown
15	ORF 15	13,129	15,057	1929	Hypothetical Protein	Currently unknown
16	ORF 16	15,050	15,289	240	Hypothetical Protein	Currently unknown
17	ORF 17	15,282	15,470	189	Hypothetical Protein	Currently unknown
18	ORF 18	15,510	16,280	771	Coil containing protein	Additional function
19	ORF 19	16,354	16,893	540	Hypothetical Protein	Currently unknown
20	ORF 20	16,908	17,279	372	Hypothetical Protein	Currently unknown
21	ORF 21	17,269	18,030	762	Polyamine aminopropyl transferase	Additional function
22	ORF 22	18,032	18,226	195	Hypothetical Protein	Currently unknown
23	ORF 23	18,333	18,881	549	Hypothetical Protein	Currently unknown
24	ORF 24	18,881	19,198	318	Hypothetical Protein	Currently unknown
25	ORF 25	19,274	19,651	378	Hypothetical Protein	Currently unknown
26	ORF 26	19,645	20,028	384	Hypothetical Protein	Currently unknown
27	ORF 27	20,218	22,155	1938	Hypothetical Protein	Currently unknown
28	ORF 28	22,250	22,486	237	Hypothetical Protein	Currently unknown
29	ORF 29	22,499	24,376	1878	Hypothetical Protein	Currently unknown
30	ORF 30	24,471	24,782	312	Hypothetical Protein	Currently unknown
31	ORF 31	24,802	25,536	735	Hypothetical Protein	Currently unknown
32	ORF 32	25,588	26,301	714	Hypothetical Protein	Currently unknown
33	ORF 33	26,306	26,560	255	Hypothetical Protein	Currently unknown
34	ORF 34	26,563	26,922	360	Hypothetical Protein	Currently unknown
35	ORF 35	26,935	27,432	498	Hypothetical Protein	Currently unknown
36	ORF 36	27,435	27,779	345	Hypothetical Protein	Currently unknown
37	ORF 37	27,763	28,176	414	Hypothetical Protein	Currently unknown
38	ORF 38	28,323	28,727	405	Hypothetical Protein	Currently unknown
39	ORF 39	28,724	29,125	402	Hypothetical Protein	Currently unknown
40	ORF 40	29,118	29,561	444	Hypothetical Protein	Currently unknown
41	ORF 41	29,569	29,982	414	Hypothetical Protein	Currently unknown
42	ORF 42	30,435	31,250	816	Hypothetical Protein	Currently unknown
43	ORF 43	31,411	31,752	342	Hypothetical Protein	Currently unknown
44	ORF 44	31,798	32,142	345	Hypothetical Protein	Currently unknown
45	ORF 45	32,300	32,740	441	Hypothetical Protein	Currently unknown
46	ORF 46	32,740	32,985	246	Hypothetical Protein	Currently unknown
47	t RNA	33,833	33,916	84	Lysine	Additional function
48	ORF 48	34,327	34,752	426	Hypothetical Protein	Currently unknown
49	ORF 49	34,752	35,234	483	Hypothetical Protein	Currently unknown
50	ORF 50	35,280	35,669	390	Hypothetical Protein	Currently unknown
51	ORF 51	35,801	36,199	399	Hypothetical Protein	Currently unknown
52	ORF 52	36,303	36,671	369	Hypothetical Protein	Currently unknown
53	ORF 53	36,819	38,903	2085	Hypothetical Protein	Currently unknown
54	ORF 54	38,906	39,238	333	Hypothetical Protein	Currently unknown
55	ORF 55	39,228	40,010	783	Hypothetical Protein	Currently unknown
56	ORF 56	40,110	40,238	129	Hypothetical Protein	Currently unknown
57	ORF 57	40,818	41,078	261	Hypothetical Protein	Currently unknown
58	ORF 58	41,161	41,376	216	Hypothetical Protein	Currently unknown
59	ORF 59	41,366	41,824	459	Hypothetical Protein	Currently unknown
60	ORF 60	41,821	42,075	255	Hypothetical Protein	Currently unknown
61	ORF 61	42,075	42,263	189	Hypothetical Protein	Currently unknown
62	ORF 62	42,244	42,627	384	Putative DNA Polymerase	DNA metabolism
63	ORF 63	42,631	42,771	141	Hypothetical Protein	Currently unknown
64	ORF 64	42,768	43,400	633	Hypothetical Protein	Currently unknown
65	ORF 65	43,511	43,933	423	Hypothetical Protein	Currently unknown
66	ORF 66	43,945	44,355	411	TM Helix protein	Additional function
67	ORF 67	44,439	44,753	315	Hypothetical Protein	Currently unknown
68	ORF 68	44,870	45,565	696	DNA Methyltransferase	DNA metabolism
69	ORF 69	45,580	46,686	1107	Par B Nuclease domain protein	Additional function
70	ORF 70	46,679	47,014	336	MazG-like family protein	Additional function
71	ORF 71	47,030	48,874	1845	Gene transfer agent portal protein	Additional function
72	ORF 72	48,938	51,307	2370	DNA Polymerase	DNA metabolism
73	ORF 73	51,312	52,214	903	Transporter protein	Additional function
74	ORF 74	52,205	53,263	1059	Hypothetical Protein	Currently unknown
75	ORF 75	53,369	53,932	564	Putative protein—Tyrosine phosphatase	Additional function
76	ORF 76	53,936	54,403	468	Hypothetical Protein	Currently unknown
77	ORF 77	54,406	56,481	2076	Pyruvate/phosphate dikinase	Additional function
78	ORF 78	56,547	56,936	390	Coil containing protein	Additional function
79	ORF 79	56,946	57,509	564	Hypothetical Protein	Currently unknown
80	ORF 80	57,499	57,738	240	Hypothetical Protein	Currently unknown
81	ORF 81	57,873	58,118	246	Hypothetical Protein	Currently unknown
82	ORF 82	58,132	60,006	1875	Terminase large subunit	Packaging
83	ORF 83	60,003	60,830	828	Coil containing protein	Additional function
84	ORF 84	60,845	61,204	360	Hypothetical Protein	Currently unknown
85	ORF 85	61,268	62,251	984	Hypothetical Protein	Currently unknown
86	ORF 86	62,460	63,014	555	Cross over junction endodeoxyribonuclease	DNA metabolism
87	ORF 87	63,016	64,062	1047	Rubredoxin-type fold protein	Additional function
88	ORF 88	64,052	64,462	411	Hypothetical Protein	Currently unknown
89	ORF 89	64,416	65,510	1095	RecA Protein	Additional function
90	ORF 90	65,543	65,902	360	Hypothetical Protein	Currently unknown
91	ORF 91	65,904	66,305	402	Putative DNA binding domain protein	Additional function
92	ORF 92	66,381	67,733	1353	Putative DNA helicase	DNA metabolism
93	ORF 93	67,846	68,808	963	Hypothetical Protein	Currently unknown
94	ORF 94	68,852	69,847	996	DNA Primase	DNA metabolism
95	ORF 95	69,840	70,445	606	Hypothetical Protein	Currently unknown
96	ORF 96	70,500	71,960	1461	DNA B-like Helicase	DNA metabolism
97	ORF 97	71,969	72,550	582	Hypothetical Protein	Currently unknown
98	ORF 98	72,560	74,215	1656	Hypothetical Protein	Currently unknown
99	ORF 99	74,371	74,922	552	Hypothetical Protein	Currently unknown
100	ORF 100	74,919	76,127	1209	ATP-dependent zinc metalloprotease	Additional function
101	ORF 101	76,336	76,872	537	TM Helix protein	Additional function
102	ORF 102	76,877	77,323	447	Hypothetical Protein	Currently unknown
103	ORF 103	77,320	77,550	231	Hypothetical Protein	Currently unknown
104	ORF 104	77,547	77,849	303	Hypothetical Protein	Currently unknown
105	ORF 105	77,842	78,384	543	Hypothetical Protein	Currently unknown
106	ORF 106	78,381	79,679	1299	Hypothetical Protein	Currently unknown
107	ORF 107	79,689	79,862	174	Hypothetical Protein	Currently unknown

**Table 2 viruses-15-00868-t002:** Comparison of genome of Vibriophage-ϕLV6 with genomes of other vibriophages against *V. harveyi* clade (reported in the literature).

	Vibriophage-ϕLV6	Siphophage-VHS1	Vibriophage-vB_VhaM_pir03	Vibriophage-VB_VhaS_PcB-1G	Vibriophage-VB_VcaS_HC	Vibriophage-Virtus
Genome Size (bp)	79,862	81,509	286,284	48,719	81,566	82,960
G+C (%)	48	46.87	43.6	43.06	47.6	47.42
AnnotatedORFs	138	125	137	80	121	127
tRNAs	1	0	0	0	0	0
ARGs	Absent	Absent	Absent	Absent	Absent	Absent
Virulence genes	Absent	1—Shrimp haemocyte agglutination	Absent	Absent	Absent	Absent
Integrase	Absent	Absent	Absent	Absent	Absent	Absent
Aquatic Animal Species	Shrimp (*P. vannamei*)	Shrimp (*P. vannamei*)	Brine shrimp (Artemia)	Fish (Zebrafish)	-	Fish (Gilt Head Sea Bream)
Country	India	Thailand	Greece	China	China	Greece
Ref.	(This study)	[59]	[38]	[32]	[60]	[39]

**Table 3 viruses-15-00868-t003:** Storage stability of vibriophage-ϕLV6 at 4 °C.

Period of Storage at 4 °C (Months)	Vibriophage Count (Log Pfu mL^−1^)
0	10.5
1	10.5
4	10.7
9	10.4
12	9.7

## Data Availability

The data generated from this study is available from the corresponding author. The whole genome sequence data of vibriophage-Lv6 was submitted to NCBI (GenBank) and is available under accession number OP918262.1. The accession details of the associated SRA, Bioproject, and Biosample are SRX11714574, PRJNA753649, and SAMN20703012, respectively.

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
