# Peer review of "Genome Characterization and Infectivity Potential of Vibriophage-ϕLV6 with Lytic Activity against Luminescent Vibrios of Penaeus vannamei Shrimp Aquaculture"

_viruses, 2023, doi:10.3390/v15040868_

Round 1
Reviewer 1 Report
The manuscript viruses-2222722 entitled “Genome characterization and infectivity potential of Vibriophage-Ï•LV6 with lytic activity against luminescent vibrios of Penaeus vannamei shrimp aquaculture” provides a comprehensive scientific results on the the isolation, genomic characterization of vib riophage-ɸLv6 and assessed it’s in vitro and in vivo lytic ability against luminescent Vibrio harveyi.
In general, the topic is of important for the readers of viruses as well as for the aquaculture scientists. The manuscript was well designed and several parameters were investigated. The “Introduction” part and “Material and Methods” part are written and well structured. However, there is some issues with the “Results and Discussion” part. I would suggest the authors to separate these two parts. Otherwise, this part is confusing. Having separate sections allows the reader to more easily understand the key findings of the research and the interpretations and implications of those findings. Moreover, the discussion must deeply and clearly discuss the results, not just report the results.
My specific comments are as below:
I couldn't find the line numbers in this manuscript, it makes it difficult to review.
1. Introduction
Page 2: “cause mortalities in P. monodon”. it must be full name as here is mentioned for the first time.
Page 2: “isolated from Penaeus monodon”. it most be abbreviated
2. Material and Methods:
Page 3: “Illumina HiSeq.” What kinds of Illumina HiSeq did you use?
Page 4: “V. harveyi host i.e., LV6” For the challenge studies. Why did you only consider “LV6” but not others? What is specific with LV6?
Page 4: “and Penaeus vannamei post-larvae”. it most be abbreviated . Apply this for whole parts of the manuscript.
Page 4: what was the water temperate, oxygen, photo-period, and feeding?
Page 4: “Control tanks were” “c” must be in lowercase not capital
3. Results and Discussion
Page 5: In Figure 1. “P.vannamei” must be italic
Page 7: for “74178 bp” add”,”. “74,178 bp consider in the whole manuscript
I think it would be better to separate and “results” and “Discussion” parts. Having separate sections allows the reader to more easily understand the key findings of the research and the interpretations and implications of those findings. The discussion must deeply discuss the results, not just report the results.
Reviewer 2 Report
This study presented the genomic analysis and infectivity of a Vibriophage-ϕLV6 against luminescent vibrios of Penaeus vannamei shrimp aquaculture. Generally, this study is interesting. However, several concerns should be addressed before it could be considered for publication.
My detail comments are listed as follows:
1.Abstract
“Transmission electron microscopy (TEM) of vibriophage-ɸLV6 revealed an icosahedral head (~73 nm) and long flexible tail (~191 nm) suggesting Siphoviridae.”
It is not accurate to use “Siphoviridae”. According to the latest classification standard of ICTV, there is a re-classification of Caudovirales. Some traditional families, such as Siphoviridae, Myoviridae, Podoviridae, have been replaced by other new families. You can change the wording to “Siphovirus”. Other relevant parts of the manuscript should also be modified.
2. Introduction
The author seems to want to emphasize the importance of Vibrio harveyi phage in aquaculture in the introduction. However, the cited arguments are far from enough. The bacteriophage groups of Vibrio and their role in bacteriophage therapy should be explained in more detail.
3.Method
The author repeatedly mentioned “the 2-step microtiter plate assay” and also cited relevant references. However, there should also be a brief description of the 2-step microtiter plate assy in this paper. In addition, the one-step growth curve is an important indicator to evaluate the bacteriophage lysis ability. I don't know why it is not reflected in this manuscript.
In “Isolation of luminescent V. harveyi hosts” part
“The luminescent colonies were isolated, purified and tested for V. harveyi based on biochemical tests [29] and used as bacterial hosts.”
What biochemical experiment? What are the steps of isolation and purification? Please describe in detail.
In “Vibriophage-ɸLV6 activity against luminescent V. harveyi host LV6 at different salt
gradients” part
Why do you choose the salinity parameter for the experiment? What does the author want to explain when measuring the salinity gradient? Please briefly describe in the method section.
4. Results and Discussion
In “Isolation of Luminescent vibrios” part
“The remaining luminescent vibrios were identified to the genus level and categorized as luminescent Vibrio spp.”
What are the criteria for identifying the genus of remaining luminescent vibrios. Please specify.
In “Isolation and lytic spectrum of Vibriophage” part
“showed clear lytic zone against luminescent V. harveyi-LV6 (Fig. 2) after single-host en-
richment and yielded pinpointed plaques with a diameter of <1 mm on single agar
method. Similarly, Misol et al. (2020) reported that vibriophage-vB-VhaM-pir03 produced
pin-hole plaques with a diameter of 0.27± 0.05 mm on V. harveyi seeded plates.”
At present, it is arbitrary to determine whether the bacteriophage is lytic or lysogenic only depending on its morphology. More should be judged from the gene level. Morphology can only be used as a part of reference. Please modify.
“The vibriophage count after purification, PEG precipitated and suspension in SM buffer was 3.42 x 1010 pfu mL-1 as determined in the single agar method. PEG precipitation increased the concentration of the vibriophage 100 times. Similar increase was reported in coliphage by PEG precipitation [32]. We have observed that there was no difference in the counts of vibriophage-ɸLV6 on single agar method and double agar overlay methods.”
First of all, the manuscript does not provide any evidence to show how pfu is calculated. The one-step growth curve is also not given. Secondly, there is no relevant description of the comparison between single agar method and double agar method in the method part of the manuscript. Please change the wording.
In “TEM Morphology of Vibriophage-ɸLV6” part
“The morphology of Vibriophage-ɸLV6 observed under Transmission Electron Microscopy (Fig. 3) revealed that the phage had an icosahedral head (~73nm) and a long filamentous non-contractile tail (~191nm), respectively suggesting that the vibriophage-ɸLV6 might belong to the family of Siphoviridae. Majority of the phages that showed lytic activity against V. harveyi were reported to be Siphoviruses [19,20,26,47-50] followed by Myoviruses [19,25,46,50,51].”
Only relying on morphological identification is outdated. It is necessary for the author to combine the information at the genome level to determine which family bacteriophages belong to. In addition, the traditional families of Caudovirales on ICTV have been replaced, and the author needs to make a more rigorous judgment on the classification of bacteriophages.
In“Multiple genome alignments“ part
“Vibriophage-ɸLv6 (NCBI Accession ID: OP918262.1) had genome sequence similarity of 79.89%, 74.97% and 74.44% to Vibriophage-vB_VpS_PG28 (NCBI Accession ID:MT735630.2), Vibriophage-V-YDF132 (NCBI Accession ID: 0N075462.1) and Vibriophage-VH2_2019 (NCBI Accession ID: MN79428.1)”
This section and the next section are logically illogical. I guess the author wants to prove that their evolutionary origins are similar, and then do comparative genomics analysis.
In “Whole genome comparison” part
“Vibriophage-ɸLV6 harboured replication, regulation, structural and packaging modules similar to vibriophage-vB_VpS_PG28 and vibriophage-VH2_2019 but lacked a lysis module.”
The author simply described the results in words, but the best way to explain this sentence is to mark the comparison of genome information of three bacteriophages in the form of figures or tables.
In “Effectiveness of vibriophage-ɸLV6 treatment at optimized MOI of 80 against a single luminescent V. harveyi-LV6” part
It is not enough for the author to show the results of this part with only one figure.
I wonder if there are parallel experiments. If yes, please put out the results of each group.
The picture only shows the effect on the host bacteria. But I haven't seen the manifestation of higher post-larvae survival.
When MOI=80, the effect of bacteriophage on the host and the survival rate of shrimp larvae should have a line chart or table to show the change results more intuitively. The author should focus on how to better display the results with pictures.
5. Others question
I don't know why the quality of figure 6.7 is poor, and the legend is not clear at all.
The footnote of figure 4 is wrong.
The conclusion part is too short and the result part is too long. Please make appropriate adjustments and deletions.
